



# Deep soil water $^{18}$O and $^{2}$H measurements preserve long term evaporation rates on China's Loess Plateau

Wei Xiang[1], Bingcheng Si[1,2], Min Li[1], Han Li[1]

[1] Key Laboratory of Agricultural Soil and Water Engineering in Arid and Semiarid Areas, Ministry of Education, North-west A&F University, Yangling, Shaanxi Province, 712100, China

[2] Department of Soil Science, University of Saskatchewan, Saskatoon, SK, Canada

*Correspondence to*: Bingcheng Si (Bing.Si@usask.ca)

**Abstract.** Knowledge about the long-term average soil evaporation, especially the ratio of evaporation to precipitation (*f*), is important for assessing the total available water resources. However, determining the long-term *f* remains technically challenging because soil evaporation is highly dynamic. Here we hypothesize that the stable isotopes ($^{2}$H and $^{18}$O) of deep soil water preserve the long-term evaporation effects on precipitation and can be used to estimate long-term *f*. Our results showed that the deep soil water (2 - 10 m) had a mean line-conditioned excess (lc-excess) less than zero (-13.1‰ to -3.8‰) at the 15 sites across China's Loess Plateau, suggesting that evaporation effects are preserved in the isotopic compositions of the deep soil water. We then estimated *f* by the new lc-excess method that combines lc-excess and the Rayleigh fractionation theory, because it does not require the initial source isotopic values of soil water, which has a distinct advantage over traditional isotope-based methods (e.g. Craig-Gordon model) that require such information a priori. The estimated *f* of the 15 sites varied from 11% to 30%, and over 60% of the variability of *f* was explained by the well-known Budyko dryness index. These data are also comparable with available annual estimates under similar climate regions of the world. Furthermore, these data represent a long-term average value because soil water tritium profile shows that deep soil water has a long residence time on the order of years to decades. Our work suggests that isotopic compositions of deep soil water can be used to calculate long-term average *f* where water flow within the unsaturated zone is piston-like flow predominantly, and the new lc-excess method provides an effective tool to estimate *f*.

## 1 Introduction

Water loss from soil by evaporation is an essential component of the terrestrial water cycle (Or and Lehmann, 2019). Esti-

mation of soil evaporation, not only the amount but also the ratios of it to precipitation ($f$) or evapotranspiration, within the

different landscapes helps understanding ecohydrological processes (Sprenger et al., 2017b;Sprenger et al., 2017a), quan-

tifying the water balance (Skrzypek et al., 2015), partitioning evapotranspiration (Anderson et al., 2017;Kool et al., 2014),

and calibrating rainfall-runoff models (Birkel et al., 2014). Because of the dynamic nature due to variation of climate, sur-

face soil water capacity, and vegetation conditions (Or and Lehmann, 2019), a more reliable assessment of soil evaporation

or $f$ needs to determine the long-term average values (Stoy et al., 2019), which remains a serious technical constraint (Kool

et al., 2014).

Many methods have been developed for measuring soil evaporation, and thus estimating $f$ by combining soil evaporation

measurements with climate record (i.e. precipitation). The field measurement methods, such as micro-lysimeters, soil heat

pulses, chambers, micro Bowen ratio energy balance, and eddy covariance (Kool et al., 2014), only estimate soil evapora-

tion at particular points and have relative short observational timescales (days to growing season) (Anderson et al., 2017).

Temporally integrating these estimates needs high-frequency continuous observations in-situ, which is time-consuming

and costly. Notwithstanding modeling is useful for estimating soil evaporation over relatively long periods, current models

are rarely capable of estimating under-canopy evaporation (Good et al., 2015) and even when they are, they tend to overes-

timate the soil evaporation and need to be validated by long-term filed measurements (Lian et al., 2018;Niu et al., 2019).

Conversely, a relatively long-term $f$ can be estimated using stable [2]H and [18]O isotopes, because they are only enriched dur-

ing soil evaporation processes (Zimmermann et al., 1966) and can integrate soil evaporation processes occurring over in-

terest of period (Jasechko et al., 2013;Allison and Barnes, 1983). The primary advantage is that it does not need continuous

field observations with high frequency and have been used to estimate $f$ in various studies (Allison and Barnes,

1983;Sprenger et al., 2017a;Hsieh et al., 1998;Mahindawansha et al., 2019). Previous studies commonly utilize the shallow

soil water [2]H / [18]O isotopes; however, obtaining long-term average $f$ requires continuous sampling effort of shallow soil

due to the strong isotopic dynamics in that part of the soil profile, which can be costly. Precipitation falling as rainfall on

the land surface undergoes a series of processes including evaporation in the canopy and the soil surface (isotope fraction-

ation), transpiration within root zone (isotope non-fractionation), and mixing in the shallow soil, and eventually infiltrates

to the deeper soil and groundwater or forms runoff down slope to a stream (Good et al., 2015). Therefore, deep soil water

isotopic compositions may serve as an ideal logger of surface soil evaporation, preserving the effects of evaporation on

precipitation over a relatively long period (DePaolo et al., 2004;Sprenger et al., 2016). Deep soil water could provide an

opportunity to estimate the long-term average $f$, but the opportunity has not been systematically explored.

Methods to determine $f$ from isotopic composition of water are generally based on the well-known Craig-Gordon model by

single-isotope system ($^{18}$O or $^{2}$H) (Craig and Gordon, 1965); however, estimates from $^{18}$O are often inconsistent with those

from $^{2}$H (Sprenger et al., 2017a;Mahindawansha et al., 2019). The dual-isotope systems, including the deuterium-excess

(d-excess, as defined by Dansgaard (1964)) and the line-conditioned excess (lc-excess, Landwehr and Coplen (2006)),

overcome the inconsistency, but are often used as a qualitative indicator of evaporation intensity rather than quantitatively.

That is, the d-excess or lc-excess of a water sample, display the degree of evaporation that the water experienced only; but

the fraction of evaporation loss to the initial water body (i.e. $f$) can not be determined by d-excess or lc-excess. Recent

works suggest that combing d-excess and the Rayleigh fractionation theory (Clark and Fritz, 1997) can estimate $f$ quantita-

tively and is shown to be more robust than the traditional Craig-Gordon method (Hu et al., 2018;Zhao et al., 2018). How-

ever, a key problem is that the d-excess is not only an indicator of evaporation but also reflects both the original source of

water vapor and of relative humidity in the source area (Masson-Delmotte et al., 2005). Conversely, lc-excess defines the

offset of a water sample to its Local Meteoric Water Line (LMWL) due to kinetic fractionation caused by evaporation only

(Sprenger et al., 2017b). But to the best of our knowledge, little is known about the performance of lc-excess on quantify-

ing $f$.

Therefore, the overall objective of this study was to estimate long-term $f$ quantitatively by combining deep soil water iso-

topic compositions and lc-excess, based on the deep soil water isotopic compositions measurement at multiple sites on

China's Loess Plateau. Specifically, we ask the following questions: (i) Can we use deep soil water isotopic compositions

to estimate soil evaporation to precipitation ratio? and (ii) How to use lc-excess in estimating evaporation to precipitation

ratio?

## 2 Materials and methods

### 2.1 Study site

China's Loess Plateau, located in northern China, is the largest loess deposit in the world, with a mean thickness of 100 m
and the maximum thickness of up to ~350 m (Zhu et al., 2018). Loess is quaternary sediment and is marked by three layers:
the upper Late Pleistocene Malan Loess, the middle Middle Pleistocene Lishi Loess, and the lower Early Pleistocene Wu-
cheng Loess (Liu, 1985). This region is characterized by a warm temperate continental monsoon climate, with the mean
annual precipitation, potential evapotranspiration (Penman, 1948), and temperature of 200-900 mm, 700-1400 mm, and
8-14℃, respectively (**Fig. 1**). The groundwater level is 50-100 m below the land surface.

### 2.2 Sampling and measurement

We obtained a total of 15 deep soil cores under different climate regions (**Fig. 1d**; **Table 1**):12 soil cores (S1-7 and S9-13)
were sampled over the period 2015-2019, and the remaining cores (S8 and S14-15) collected from the literature where
deep soil water isotope information was available. The vegetation of the sampling sites is farmland and grassland. The
farmland was characterized by long-term (more than 100-year-old) rain-fed winter wheat or summer maize, and the grass-
lands were converted from farmlands. All of the above soil sites are located at a very flat landscape where runoff is often
negligible.

In order to assess the vertical distribution of soil water isotopic composition, a hand auger with stem extension was used at
each site to obtain samples at an interval of 0.2 m to a depth of 9-10 m. Each soil sample was stored in a 250 mL plastic
bottle sealed with parafilm, transported to the laboratory and stored in the refrigerator at -20 °C prior to soil water extrac-
tion. Further, to assess the temporal variability of soil water isotopic composition, we collected eight 5-m soil cores be-
neath a 13-year-old apple orchard (converted from farmland, < 200 m away from the farmland) at site S11 during
2015-2016 (collected on 2015/7/12, 2015/8/19, 2016/3/20, 2016/5/30, 2016/7/3, 2016/7/30, 2016/8/30, and 2016/10/15).





Soil water was extracted via the cryogenic vacuum distillation method (Orlowski et al., 2016), and the water extraction and collection efficiencies both ranged from 98% to 102% for each sample. The extracted water was then stored in a 10 mL

glass bottle and kept in the refrigerator at 4 °C before stable isotope analysis. The stable isotopic compositions ($^{18}$O and $^{2}$H) of soil water samples were determined with an isotopic liquid water analyzer (LGR LIWA 45EP, USA), and the isotopic values are presented using the delta notation (in ‰):

$$\delta_{Sample} = \left(\frac{R_{Sample}}{R_{VSMOW}} - 1\right) \times 1000 \qquad (1)$$

where the $R_{sample}$ and $R_{VSMOW}$ are the $^{2}$H/$^{1}$H ($^{18}$O/$^{16}$O) ratio of sample and Vienna Standard Mean Ocean Water (VSMOW),

respectively. The precision of the isotopic measurements was 1.0‰ and 0.2‰ for $\delta^{2}$H and $\delta^{18}$O, respectively.

**2.3 Theory and methods**

**2.3.1 Lc-excess**

Landwehr and Coplen (2006) specified that the offset of a water sample from the LMWL is the lc-excess:

$$lc - excess = \delta^{2}H - a\delta^{18}O - b \qquad (2)$$

where the $a$ and $b$ are the slope and intercept of LMWL, respectively. Precipitation samples that have undergone minimal to no evaporation prior to sample collection would have, by definition, an lc-excess of zero. A negative lc-excess value that is smaller than one negative standard deviation of the precipitation suggests that a water sample has undergone some evaporative isotopic enrichment. The smaller the lc-excess value, the stronger the influence of evaporation. Therefore, we used the lc-excess as a qualitative indicator of the evaporation to precipitation ratio.

We obtained long-term precipitation isotopic data from the International Atomic Energy Agency (IAEA) Global Network of Isotopes in Precipitation from five stations located in Zhengzhou, Taiyuan, Baotou, Xi'an, and Lanzhou (**Fig. 1d**). Based on the $\delta^{2}$H and $\delta^{18}$O of all monthly precipitation samples (n = 212), the regional LMWL was determined using the least square regression weighted by precipitation amount: $\delta^{2}$H = 6.89 (0.15) $\delta^{18}$O - 0.16 (1.23), $R^{2}$ = 0.91, which was used as the reference line for all soil sites.



### 2.3.2 Estimation of evaporation to precipitation ratio

Precipitation water entering the soil may be partially evaporated gradually, and the water remaining in the soil will thus be enriched in heavy isotopes ($^{18}O$ and $^{2}H$) by evaporation fractionation (Sprenger et al., 2016). Assuming that soil water evaporation follows the Rayleigh fractionation theory (Clark and Fritz, 1997), the evaporation to precipitation ratio $f$ of a sampled soil water after evaporation can be written as:

$$R_s = R_0(1-f)^{(\alpha-1)} \tag{3}$$

where $\alpha$ is a fractionation factor, and $R_s$ and $R_0$ are the $^{2}H/^{1}H$ (or $^{18}O/^{16}O$) ratio of the evaporated water sample and its initial water source, respectively. Based on Eqs. (1) and (3), we used $\delta_{(\bullet)}$ to replace the $R_{(\bullet)}$, and then Eq. (3) can be written as:

$$\delta_0 = (\delta_s + 1000)(1-f)^{(1-\alpha)} - 1000 \tag{4}$$

where $\delta_s$ and $\delta_0$ are the isotopic values of the evaporated water sample and the initial water source, respectively. Combining Eqs. (2) with (4), we obtain:

$$lc_0 = \frac{[\delta^2H_s+1000]}{(1-f)^{[\alpha(^2H)-1]}} - \frac{a(\delta^{18}O_s+1000)}{(1-f)^{[\alpha(^{18}O)-1]}} + 1000(a-1) - b \tag{5}$$

where $lc_0$ is the lc-excess value of the initial water source. $\delta^2H_s$ and $\delta^{18}O_s$ are the isotopic compositions of an evaporated water sample. $\alpha_{(\bullet)}$ is a fractionation factor that reflects both the equilibrium fractionation ($\alpha^*_{(\bullet)}$) and the kinetic enrichment ($\varepsilon_{k(\bullet)}$) factors ($\alpha_{(\bullet)} = \frac{1}{\alpha^*_{(\bullet)}+\varepsilon_{k(\bullet)}}$) (Clark and Fritz, 1997). The $\alpha^*_{(\bullet)}$ is a function of temperature ($T$) in degrees Kelvin (Eqs. (6) and (7)) (Gonfiantini, 1986; Majoube, 1971), and the $\varepsilon_{k(\bullet)}$ is related to relative humidity ($Rh$) (Eqs. (8) and (9)) (Horita et al., 2008; Benettin et al., 2018):

$$ln\left[\alpha^*_{(^2H)}\right] = \frac{24.844}{T^2}(10^3) - \frac{76.248}{T} + 52.612(10^{-3}) \tag{6}$$

$$ln\left[\alpha^*_{(^{18}O)}\right] = \frac{1.137}{T^2}(10^3) - \frac{0.4156}{T} - 2.0667(10^{-3}) \tag{7}$$

$$\varepsilon_{k(^2H)} = n(1-Rh)(1-0.9755) \tag{8}$$

$$\varepsilon_{k(^{18}O)} = n(1-Rh)(1-0.9723) \tag{9}$$

where the parameter $n$ accounts for the aerodynamic regime above the evaporating liquid-vapor interface, which ranges from 0.5 (saturated soil condition) to 1.0 (very dry soil condition). We set n to 0.75 because the evaporating soil layer was



expected to the alternation of saturating and drying over time (Benettin et al., 2018). We estimated the monthly $\alpha^*_{(\cdot)}$ and

$\varepsilon_{k(\cdot)}$ using monthly temperature and relative humidity, and then the annual values are estimated by summation of the

monthly values weighted by the monthly temperature and relative humidity. If $\alpha^*_{(\cdot)}$ and $\varepsilon_{k(\cdot)}$ are calculated, the long-term

mean annual $\alpha_{(\cdot)}$ can be determined. For each soil site, the long-term (1981-2010) mean monthly temperature and relative

humidity data were obtained from the nearby meteorological station (**Fig. 1**).

The $lc_0$ was set to zero because we assumed that soil water at our study sites only originates from the local precipitation

over a long time (Cheng et al., 2014). If the isotopic composition of soil water ($\delta^2H_s$ and $\delta^{18}O_s$), the slope ($a$) and intercept

($b$) of LWML, and climate parameters ($T$ and $Rh$) are all known, the $f$ can be calculated from Eqs. (5) to (9). In the Eq. (5),

the uncertainties resulting from the calculation of $\alpha_{(\cdot)}$ can be ignored because the multi-year mean monthly temperature and

relative humidity are used (Zhao et al., 2018). Therefore, the main sources of uncertainty for $f$ ($S_f$) are from the measured

isotopic compositions of deep soil water (2-10 m), and slope ($a$) and intercept ($b$) of LMWL. If $x = \delta^2H_s$ and $y = \delta^{18}O_s$, $S_f$

can be estimated based on the first-order perturbation analysis:

$$\left(S_f\right)^2 = \left(\frac{\partial f}{\partial x}\right)^2 \cdot (S_x)^2 + \left(\frac{\partial f}{\partial y}\right)^2 \cdot \left(S_y\right)^2 + \left(\frac{\partial f}{\partial a}\right)^2 \cdot (S_a)^2 + \left(\frac{\partial f}{\partial b}\right)^2 \cdot (S_b)^2 \qquad (10)$$

where the $S_{(\cdot)}$ represents the standard error of the relevant variable and the $\frac{\partial f}{\partial (\cdot)}$ represents the partial derivative of $f$ with

respect to the relevant variable.

## 3 Results

### 3.1 Stable isotopic compositions of soil water

**Figure 2** shows the vertical fluctuations and temporal variations of soil water isotopic compositions at different depths at

site S11. Generally, the $\delta^2H$ and $\delta^{18}O$ of soil water changed with depth, but shallow soil (0-2 m) displayed more fluctua-

tions along with depth than deep soil (> 2 m). For example, the $\delta^2H$ value varied from -77.2 ‰ to -23.1 ‰ in the top 2 m,

but only from -71.5 ‰ to -64.8 ‰ at depth >2 m. Similarly, the temporal variability of soil water isotope composition at

S11 was the largest in the surface soil and gradually weakened with increasing depth and became stable below 2.0 m.

Therefore, both the vertical fluctuations and temporal variability at S11 suggested that soil water isotopic compositions were stabilized below 2.0 m. Further, despite of differences among the sites, all the 15 deep profiles showed that soil water isotopic compositions were stabilized with depth below 2 m (**Fig. 3**).

The mean lc-excess values of deep soil water (> 2 m) for the 15 sites ranged from -13.1‰ to -3.8‰, and was significantly less than that of precipitation (p<0.05) (**Fig. 4**). For all sites, the lc-excess difference between deep soil water and precipi-

tation were larger than that of the measurement error (1.02‰). Thus, these suggested that deep soil water isotopes preserve obvious isotope fractionation signatures. Furthermore, deep soil water lc-excess values differed significantly among sites ($p < 0.05$). This reflects the difference in evaporation intensity among sites. For instance, the S8 had the smallest mean lc-excess (-13.1‰) while S6 had the largest value (-3.8‰), suggesting that evaporation intensity at S8 was stronger than that at S6.

**3.2 Estimation of evaporation to precipitation ratio**

We used deep soil water isotopic compositions and estimated evaporation to precipitation ratio ($f$) for each site using Eq. (5) and the associated uncertainty calculated using Eq. (12) (**Fig. 5a**). The resulting $f$ differed among sites and varied from 11% to 30%, with an average of 21% and a standard deviation of 6%. The uncertainty is relatively small, being within the range of 2.5% - 8.7%. Moreover, the $f$ values had a strong negative linear relationship with the deep soil water lc-excess ($f$

= -2.086lc-excess + 4.086, $R^2 = 0.81$, $p < 0.001$). This is reasonable because the lc-excess is a qualitative index of evaporation intensity, and the more intense evaporation the water experiences, the lower the lc-excess and the larger the $f$.

Both the $f$ and lc-excess were poorly correlated to the mean annual potential evapotranspiration ($R^2 = 0.04$ for lc-excess; $R^2 = 0.01$ for $f$) (**Fig. 5b**). But they were strongly statistically significantly correlated with the mean annual precipitation ($R^2 = 0.68$ and $p < 0.05$ for lc-excess; $R^2 = 0.67$ and $p < 0.001$ for $f$) (**Fig. 5c**) and the Budyko dryness index ($R^2 = 0.74$ and

$p < 0.001$ for lc-excess; $R^2 = 0.63$ $p < 0.001$ for $f$) (**Fig. 5d**). This indicates that evaporation intensity at our study sites is closely correlated to the precipitation and dryness index rather than the potential evapotranspiration.



## 4 Discussion

### 4.1 Why deep soil water isotope preserves soil evaporation effects?

Soil evaporation is a major factor for isotope enrichment of infiltration precipitation; however, it exponentially decreased
with soil depth (Zimmermann et al., 1966;Allison and Barnes, 1983), resulting in obvious isotope fractionation signatures
within a few centimeters to dozens of centimeters of shallow soil layers only (Sprenger et al., 2016). Here, we showed that
deep soil water isotopic compositions (2 - 10 m) from all sites on China's Loess Plateau indeed preserve the obvious iso-
tope fractionation signatures, as evidenced from the negative lc-excess values (**Fig. 4**). However, the isotope fractionation
signatures of deep soil water cannot be attributed directly to surface soil evaporation at the current condition because of the
thick soil depths. Nevertheless, this is consistent with many studies that also detected isotopic fractionation signals in deep
soil (DePaolo et al., 2004;Evaristo et al., 2016;Allison and Hughes, 1983;Fontes et al., 1986). For instance, Evaristo et al.
(2016) showed that evaporative isotopic enrichment either did not systematically decrease with depth or that evaporation
was restricted to the top 10 cm and transported vertically with depth.

What are the mechanisms that lead to the relative $^2$H and $^{18}$O enriched in deep soil water? As is well known, precipitation
infiltration and subsequent downward percolation within soils is often described as two end-member scenarios: piston flow
and preferential flow (Gazis and Feng, 2004). In piston flow mode, water from more recent precipitation forces the older
soil water to flow down. Under such scenario, deep soil water most likely originated from a mixture of precipitation waters
that underwent variable degrees of evaporation within the shallow layers and then moved down to these depths and thus
isotopic fractionation signal in deep soil is possible. In preferential flow mode, water bypass soil matrix and directly infil-
trated into the deep soil. Thus, deep soil water will not show obvious isotopic fractionation signals. Given that the nature of
these two flows coexist within filed soils (Lin, 2010) and their proportions (piston flow vs preferential flow) depending on
local soil characteristics (Gazis and Feng, 2004), deep soil water would show obvious isotopic fractionation signals when
piston flow is not negligible.

On China's Loess Plateau, numerous works showed that piston flow is the dominant flow within the deep unsaturated zone
(Lin and Wei, 2006;Zhang et al., 2017;Zheng et al., 2017;Yang and Fu, 2017;Huang et al., 2019;Li et al., 2019). Thus, deep

soil water would originate from above water that experienced soil evaporation and then moved into these depths by piston flow, and thus the observed isotopic fractionation within deep soil would be reasonable and can be attributed to historic evaporation. Therefore, the soil profile can be divided into mixing and stabilized zones (**Fig. 2c**). Isotopic compositions of mixing zone or deep soils may reflect the integrated influence of evaporation on precipitation water. Therefore, deep soil

water isotopic compositions may indeed be used for the estimation of evaporation to precipitation ratio.

**4.2 What is the timescale of our f estimates based on deep soil water isotopes?**

The stable isotopic compositions of soil water stabilized below approximately 2.0 m, for all of the 15 sites (**Fig. 2 and 3**). Precipitation and soil evaporation are two main factors to the variation of soil water isotopes; however, soil evaporation is limited to a few to dozens of centimeters only (Cheng et al., 2014) and precipitation infiltration depth rarely exceeds two

meters at our study area (Jin et al., 2018;Zhao et al., 2019). Therefore, the invariable isotope compositions of deep soil water indicate that deep soil water is free of seasonal variation in evaporation and precipitation infiltration (Thomas et al., 2013;Sprenger et al., 2016;Cheng et al., 2014). This is consistent with previous isotope studies (Wan and Liu, 2016;Yang and Fu, 2017) and is supported by the observation of soil water dynamics in this region (Zhao et al., 2019). Moreover, the reduced variations of stable isotopes in deep soils can be attributed to the mixing processes in the deep unsaturated zone

since it has the long residence time (DePaolo et al., 2004;Thomas et al., 2013).

A clearly-identifiable tritium peak corresponding to a 1963 precipitation input peak is located at the depth of 6.1 at S11 (**Fig. 2c**) and 9.8 m at S7 (Li and Si, 2018). Thus, the pore water velocities were 11 and 17 cm yr$^{-1}$ at S11 and S7, respectively. This suggests that that deep soil water (2 - 10 m) would have a relatively long residence time that ranges from 12 to 90 years. Thus, the *f* obtained from the average deep soil water isotopic compositions had a mean residence time of about

51 years.

The time scales considered in traditional field evaporation measurement (e.g. micro-lysimeter, and soil heat pulse) range from minutes to growing seasons (Kool et al., 2014;Anderson et al., 2017). Annual evaporation may be obtained from Eddy covariance technology, but those measurements are only available over a time span of ~ 10 years, at only dozens of sites across the world (Scott and Biederman, 2017;Gu et al., 2018). Evaporation obtained from the isotopes of groundwater

or lake water may also have long residence time like deep soil water (Jasechko et al., 2013;Zhao et al., 2018), but there are cases where the expression of isotopic fractionation signature in groundwater is modulated by the variable connectivity between mobile and immobile soil water pools (Good et al., 2015), and groundwater may be much older, and could well represent conditions far different from the current surface vegetation and climate conditions (Jasechko et al., 2017). Therefore, relative to groundwater, deep soil water is more representative of evaporation.

Our work shows that deep soil water isotopic compositions offer an effective tool to estimate the long-term average $f$, complementing other field measurements. However, we identified deep soil as soil at a depth greater than 2.0 m in this study. Sprenger et al. (2016) reported a depth of 0.3 m in the temperate regions, 0.5 m in the Mediterranean climate, and 3.0 m in the arid regions. Deep soil is common in many regions of the world (Xu and Liu, 2017), especially in arid and semiarid regions; its water isotopic compositions may become more readily available as isotope analyzers become more

accessible (Sprenger et al., 2015). Therefore, deep isotopic compositions of soil water have the potential to be widely used to estimate long-term $f$.

   Additionally, once the long-term $f$ values were estimated, the long-term transpiration to evapotranspiration fraction can be estimated with the help of measurement methods (e.g. water balance, eddy covariance, and sap flow) (Kool et al., 2014). The information can help the state-of-art models to improve the accuracy of evapotranspiration partitioning (Lian et al.,

2018;Niu et al., 2019).

**4.3 Is it reliable to use deep soil water isotopes to estimate f?**

   Our results showed that the $f$ values of the 15 sites ranged between 11% - 30%, with an average of 21% and a standard deviation of 6%. Previous metadata analysis (while data reported by Schlesinger and Jasechko (2014) as transpiration to evapotranspiration ratio, we modified these values and reported as evaporation to precipitation ratio (f); **Fig. 6a**) showed

that $f$ varied with the mean annual precipitation and had a wide range (9-79%) on the global scale. Our estimates fall within the range of that under similar precipitation region (350-800 mm yr⁻¹). In particular, their average values did not differ significantly with ours (p>0.05) and are thus comparable to our estimates (21 ± 6% vs 33 ± 15%; **Fig. 6b-c**). We understand that the data presented by Schlesinger and Jasechko (2014) collected from different methods and that the temporal

and spatial scales over which $f$ is estimated differ among the methods (Sutanto et al., 2014;Kool et al., 2014;Anderson et

al., 2017).

Additionally, we find that $f$ is poorly correlated to the potential evapotranspiration but increased with the increase in the

well-known Budyko dryness index while decreased with the increase in average annual precipitation (**Fig. 5**). Our result is

consistent with Hsieh et al. (1998), who used shallow soil water isotopic compositions to investigate evaporation along an

arid to humid transect in Hawaii. However, our result differs from the results of Sprenger et al. (2017b), who investigated

evaporation in a boreal catchment in the Scottish Highlands and showed that evaporation is positively correlated with the

potential evapotranspiration. One possible explanation for the difference is that the energy is limited for water evaporated

from soil under the wetter and colder climate condition (Budyko dryness index < 1), while our sites are more water-limited

(Budyko dryness index > 1) (Roderick and Farquhar, 2011;Good et al., 2017).

### 4.4 The advantages of using lc-excess to quantitatively estimate f

We presented a new method—coupling $f$ with the qualitative indicator of evaporation intensity (lc-excess)—based on the

Rayleigh fractionation theory, which takes advantage of isotopic fractionation effect between precipitation and soil water

to estimate $f$. Comparing with the well-known Craig-Gordon method (Craig and Gordon, 1965) and the d-excess method

(Hu et al., 2018;Zhao et al., 2018), our new method has two advantages. First, it does not require the actual stable isotopic

values of the initial water source, which is an essential parameter for the other two methods. Source water isotopic compo-

sitions are often difficult to determine in the natural water cycle (Skrzypek et al., 2015;Bowen et al., 2018), thus, evapora-

tion estimates remain an uncertainty. Locally, deep soil water only originates from local precipitation when there is no ir-

rigation (Cheng et al., 2014), and thus we can assume that the lc-excess of the initial water is zero according to the original

definition (Landwehr and Coplen, 2006). This simple assumption—the foundation for our new method and valid for most

regions around the globe—removes the requirement of the difficult-to-obtain isotopic composition of the initial water

source. As a result, our method only needs the parameters (slope and intercept) of LMWL, which can be determined using

precipitation isotope data at the interest site by monitoring, GNIP (Global Network of Isotopes in Precipitation), and pre-

cipitation isoscape (Putman and Bowen, 2019). Other isotopes-based method requires also the isotopic composition of the

initial water source, which are rarely available in applications, precluding them from their wide applications around the globe.

Another advantage is that our new method links the quantitative $f$ with lc-excess. As the lc-excess is a more representative indicator of evaporation than the single isotope system ($^{8}$O or $^{2}$H) and the d-excess (Sprenger et al., 2017b;Masson-Delmotte et al., 2005), evaporation estimates from the new method should be more accurate. Moreover, similar to the d-excess method (Hu et al., 2018), the new method combines $^{2}$H and $^{18}$O, avoiding the inconsistency between the separate implementation of $^{2}$H and $^{18}$O in the Craig-Gordon method (Sprenger et al., 2017a;Mahindawansha et al., 2019). Overall, $f$ estimation using the new method is more robust than the remaining two methods and would be widely applicable.

## 5 Conclusions

We obtained deep isotope profiles (up to 10 m) at 15 sites on China' Loess Plateau to validate the possibility of the long-term evaporation to precipitation ratio estimation using deep soil water isotopic compositions (2-10 m). First, we present a novel method to estimate $f$ with lc-excess, which, unlike the other commonly-used isotope methods, does not require the difficult-to-obtain isotopic composition of the initial water source. Second, the $f$ estimated by the new method and deep soil water isotopic compositions varies among sites, ranging from 11% to 30%, with an average of 21% and a standard deviation of 6%. These values represent a long-term average value because deep soil water has a long residence time (years to decades). They are comparable with the previous estimates at annual scale under similar climate regions of the world. Additionally, over 60% of the variability is explained by the well-known Budyko dryness index while it is poorly correlated to the potential evapotranspiration, possibly because the water supply is the first limit under such a dry climate. The proposed method could improve the $f$ estimation for regional and global water balance and apportioning precipitation water into evaporation and other hydrological processes.

**Author contribution**

W. Xiang and B.C. Si designed the research, prepared and interpreted the data, and wrote the manuscript. M. Li and H. Li

offered constructive suggestions for the manuscript. W. Xiang and H. Li conducted the fieldwork.

**Competing interests**

The authors declare that they have no conflict of interest.

**Acknowledgments**

This work was jointly funded by the Natural Science Foundation of China (41630860, 41601222, and 41877017), the

Fundamental Research Funds for the Central Universities (2452017317), and the 111 Project (No. B12007). The authors

thank Qifan Wu, Ze Tao, Keyu Liu, and Wenjie Wu for their help with the soil sampling and isotope analysis and also the

editor and reviewers for their valuable comments and suggestions. We greatly appreciate Jingjing Jin for her great support

and assistance to the instrument used in this experiment. We thank Jeffrey McDonnell for useful comments on an earlier

version of this paper.

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



**Figures**

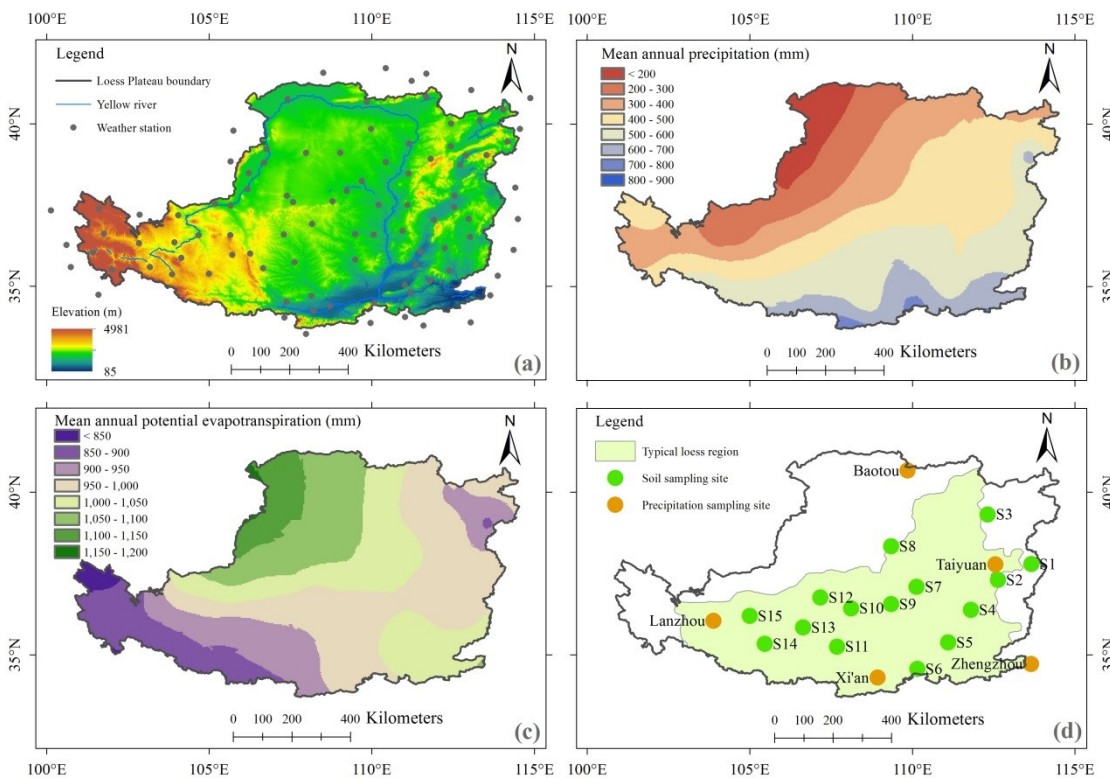

**Figure 1 Spatial patterns of climate and the distribution of the observation sites on China' Loess Plateau. (a) Elevation and weather station; (b) Mean annual precipitation; (c) Mean annual potential evapotranspiration; (d) Soil and precipitation sampling sites.**




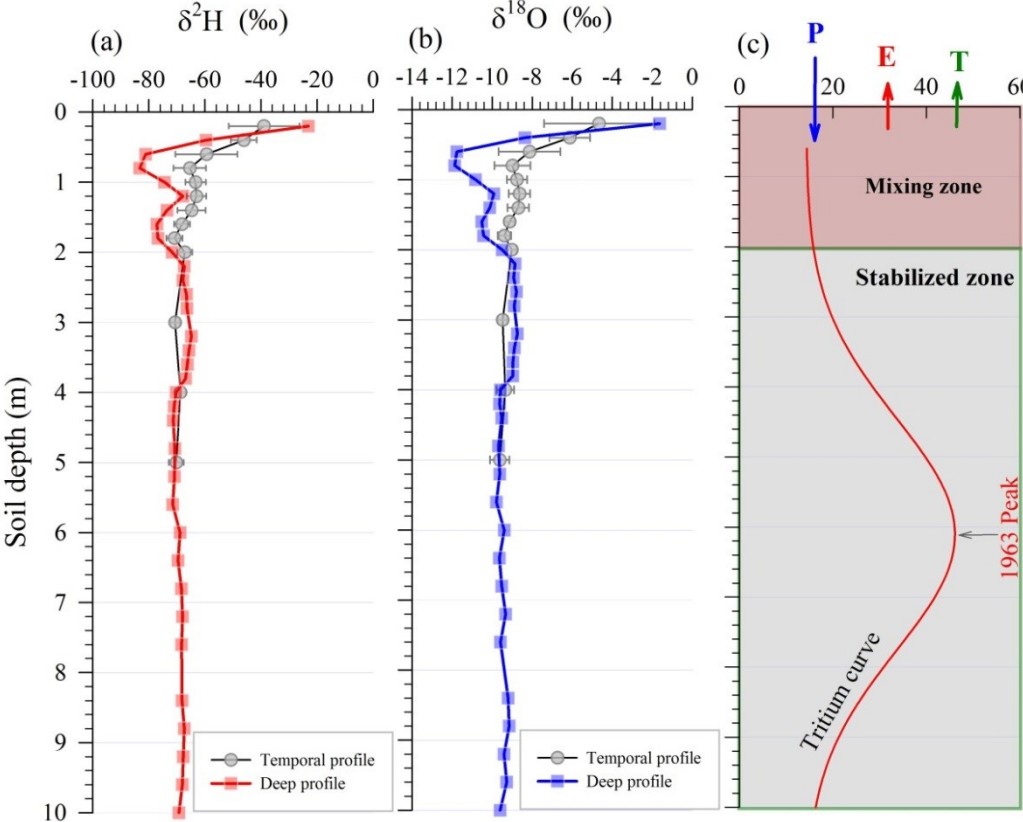

**Figure 2 Vertical and temporal distributions of δ²H (a) and δ¹⁸O (b) in soil water at S11 site. The cycle and error bars in (a) and (b) represent mean and one standard deviation from eight repeated measurements between 2015 and 2016. Figure (c) shows that soil profile can be divided into two regions: mixing zone and stabilized zone. Mixing zone represents the top zone where evapo-ration (E), transpiration (T) and precipitation infiltration (P) occur, and where the evaporation-affected water mixes with new infiltrated precipitation water. A large proportion of the mixed water returns to the atmosphere through evapotranspiration (ET = E + T) while a small proportion moves downward into the stabilized zone. The stabilized zone represents the subsurface zone where soil water isotopes no longer exhibit seasonality and are thus depth-invariant. Soil water tritium curve in (d) obtained from the same site is adapted after Li et al. (2018).**

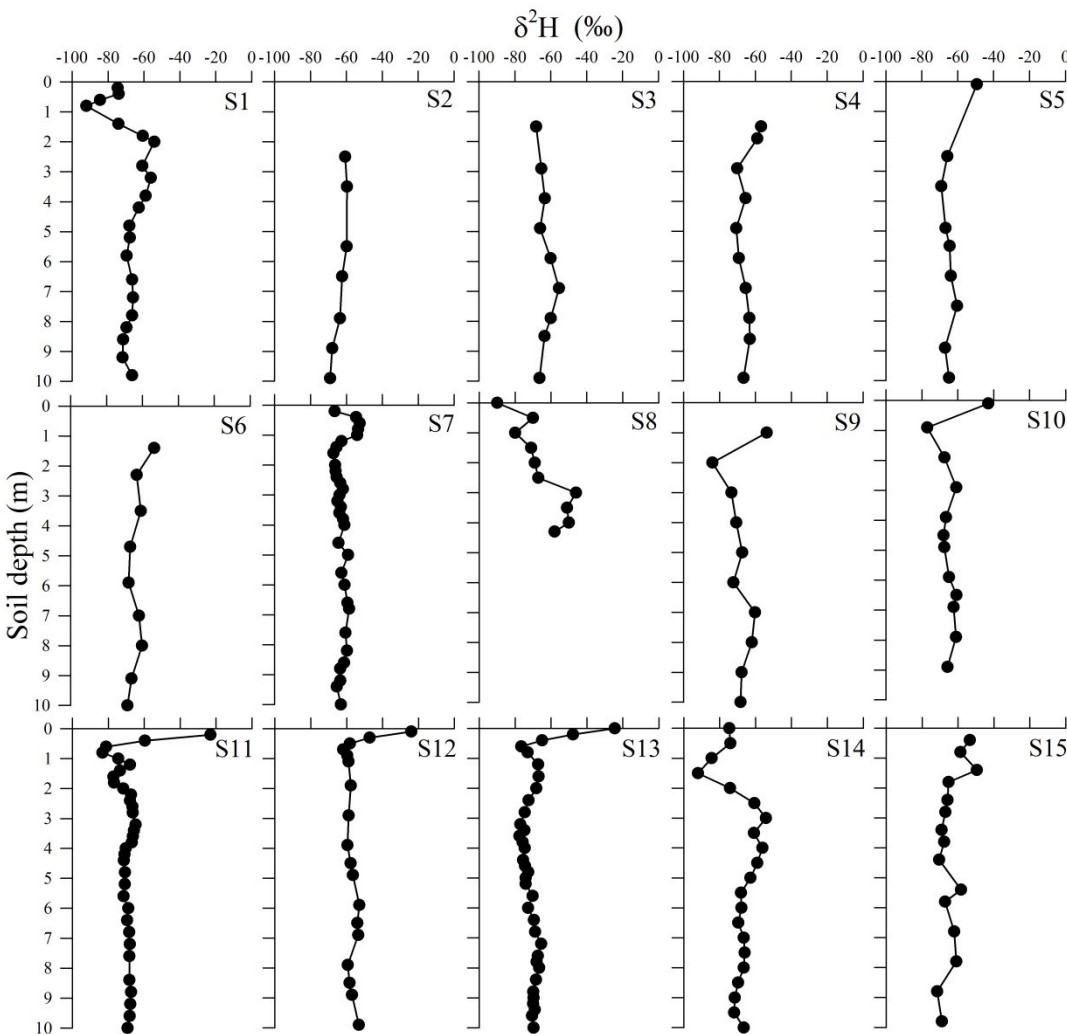

**Figure 3 The depth profiles of soil water $\delta^2H$ at 15 sites (S1-15) on China's Loess Plateau. Because $\delta^2H$ and $\delta^{18}O$ strongly covary, here we take $\delta^2H$ as an example.**



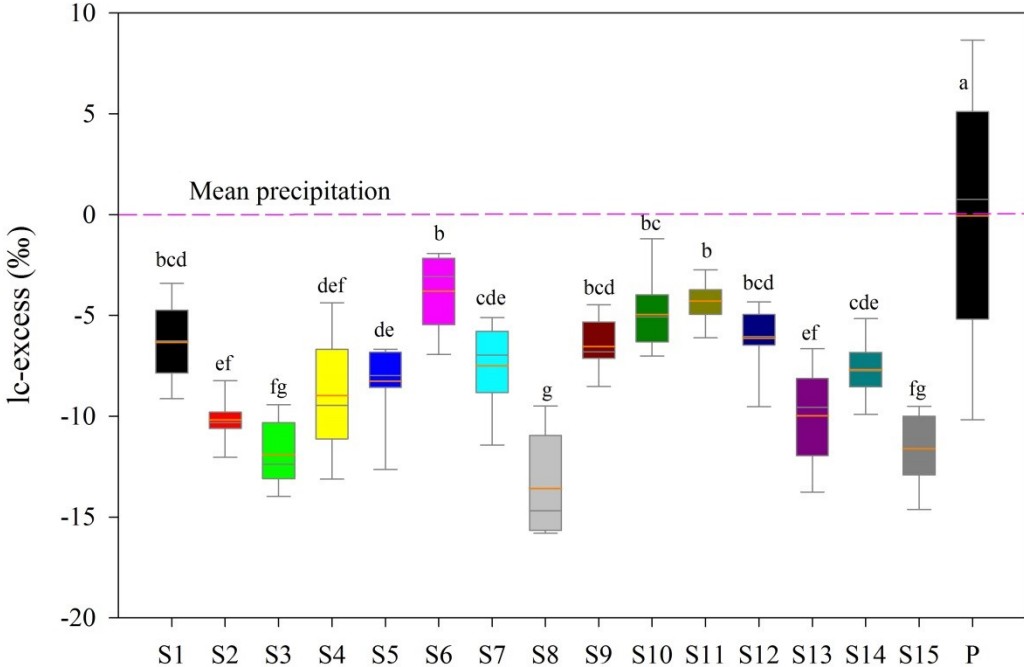

**Figure 4 Boxplots of the line-conditioned excess (lc-excess) of precipitation (P) and deep soil water (2 - 10 m) at the 15 sites (S1-15) on China's Loess Plateau. The dashed line represents the mean lc-excess of local precipitation. The boxplots show 10th, 25th, 50th, 75th, and 90th of lc-excess. The bold orange line stands the average value. Different letters next to the boxplots indicate significant differences according to the post hoc Tukey test (α=0.05).**


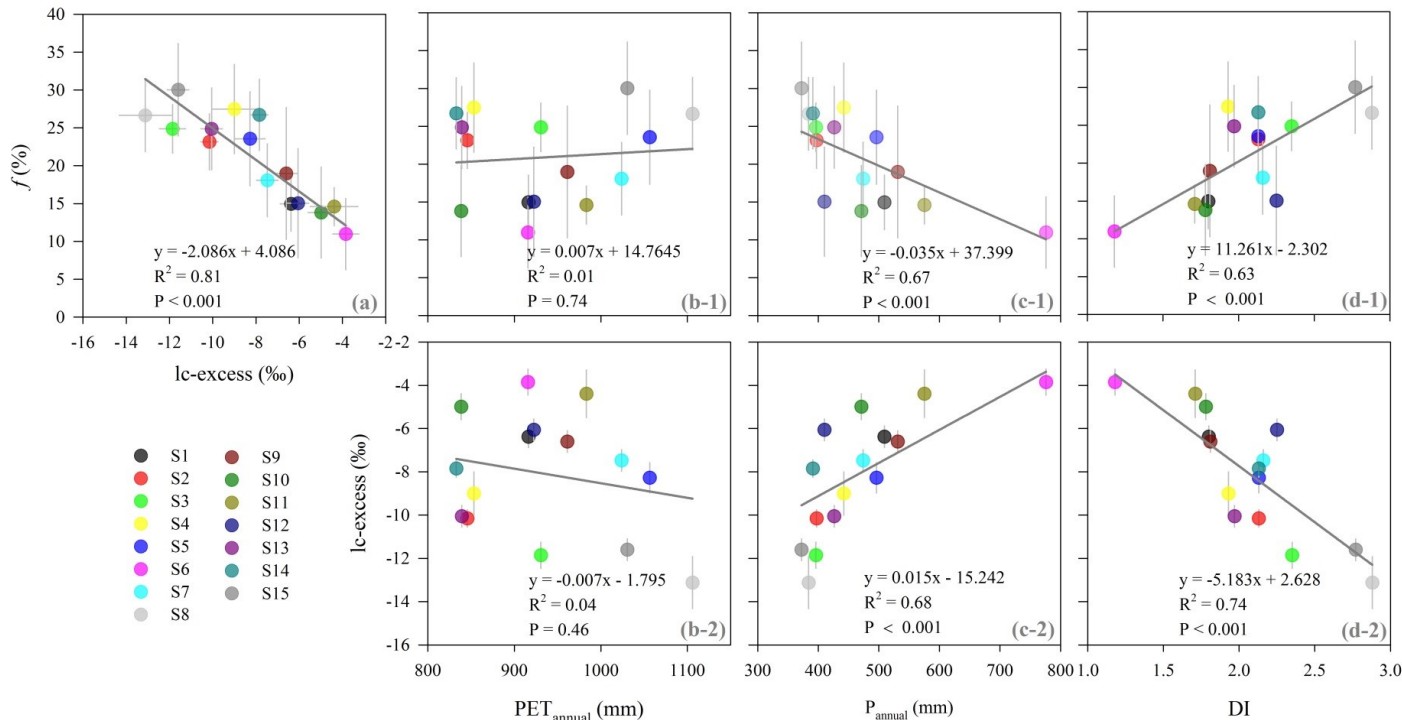

488

**Figure 5 The relationship between evaporation to precipitation ratio (*f*) and deep soil water (2 - 10 m) line-conditioned excess (lc-excess) of 15 sites (S1-15) on China's Loess Plateau (a), and their relation to the annual potential evapotranspiration (PET$_{annual}$), precipitation (P$_{annual}$), and the Budyko dryness index (DI = PET$_{annual}$ / P$_{annual}$), respectively (b-d). The gray error bars represent one standard error for each relevant variable.**





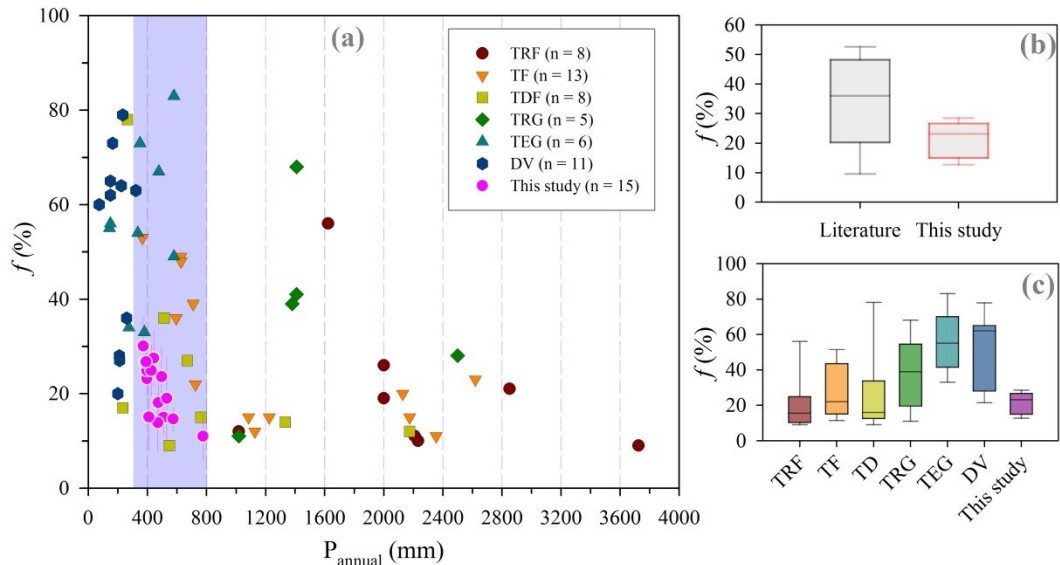

**Figure 6 The evaporation to precipitation ratio (*f*) of this study (farmland or grassland) and previous studies under different vegetation in other climatic zones of the world (data reported by Schlesinger and Jasechko (2014) and here we modified it). (a) The relationship of f between mean annual precipitation (P$_{annual}$), and the shadow in the (a) presents the P$_{annual}$ range that our estimates coverage (350-800 mm). (b) Comparison of this study with the similar climatic region (shadow coverage). (c) Comparison of this study with that under different vegetation in other climatic zones of the world. The boxplots in (b)-(c) show 10th, 25th, 50th, 75th, and 90th of the f. In (a), the vertical error bars represent the uncertainties, and TRF, TF, TD, TRG, TEG, DV stands tropical rainforest, temperate forest, temperate deciduous forest, tropical grassland, temperate grassland, and desert vegetation, respectively.**

495

500





**Tables**

**Table 1 General information about soil sampling sites on China's Loess Plateau.**

| Site | Locations | Latitude | Longitude | Elevation (m) | P (mm) | T (℃) | Rh (%) | PET (mm) | Land use | Soil depth (m) | Data source |
|---|---|---|---|---|---|---|---|---|---|---|---|
| S1 | Pingding | 37.79 | 113.66 | 850 | 509 | 10.9 | 55 | 916 | Grassland | 10 | This study |
| S2 | Taigu | 37.30 | 112.63 | 1337 | 397 | 10.4 | 58 | 846 | Farmland | 10 | This study |
| S3 | Suozhou | 39.32 | 112.32 | 1218 | 396 | 7.4 | 54 | 931 | Farmland | 10 | This study |
| S4 | Hongtong | 36.38 | 111.80 | 1004 | 442 | 12.7 | 63 | 853 | Farmland | 10 | This study |
| S5 | Wenxi | 35.38 | 111.10 | 640 | 496 | 12.9 | 65 | 1056 | Farmland | 10 | This study |
| S6 | Qinyu | 34.57 | 110.15 | 509 | 776 | 6.5 | 62 | 916 | Farmland | 10 | This study |
| S7 | Qingjian | 37.09 | 110.13 | 991 | 474 | 9.9 | 59 | 1024 | Grassland | 10 | This study |
| S8 | Yulin | 38.34 | 109.35 | 1126 | 384 | 8.8 | 54 | 1106 | - | 4.3 | Chen et al. (2012) |
| S9 | Yan'an | 36.56 | 109.34 | 1102 | 531 | 9.2 | 65 | 961 | Grassland | 10 | This study |
| S10 | Huachi | 36.43 | 108.12 | 1598 | 471 | 8.7 | 62 | 838 | Farmland | 10 | This study |
| S11 | Changwu | 35.25 | 107.68 | 1220 | 575 | 9.4 | 69 | 983 | Farmland | 10 | This study |
| S12 | Huanxian | 36.76 | 107.17 | 1415 | 410 | 9.2 | 59 | 923 | Farmland | 9 | This study |
| S13 | Pengyang | 35.84 | 106.63 | 1575 | 426 | 6.9 | 61 | 839 | Farmland | 10 | This study |
| S14 | Tongwei | 35.34 | 105.46 | 1754 | 391 | 7.2 | 70 | 833 | Grassland | 10 | Tan et al. (2017) |
| S15 | Huining | 36.19 | 105.00 | 1803 | 372 | 7.3 | 63 | 1030 | Grassland | 10 | Lin (2017) |

**P, T, and Rh are the long-tern (1981-2010) mean annual precipitation, temperature, and relative humidity, respectively. The potential evapotranspiration (PET) calculated using the Penman-Monteith formula (Penman, 1948).**