# Peer review of "Deep soil water 18O and 2H measurements preserve long term evaporation rates on China's Loess Plateau"

_Hydrology and Earth System Sciences, 2019_

## Referee Comment (RC1) · Anonymous Referee #1 · 11 Feb 2020

This paper investigated the stable isotopic compositions of soil water and used it to estimate the ratio of evaporation to precipitation at the 15 sites across China's Loess Plateau. The general idea of the paper is interesting and the general approach seems sound. However, the selection and processing of data sources make it difficult to evaluate the validity of the results. There are only four sample sites for precipitation across this area and its variability is so large, which makes it unconvincing.

Generally, the introduction and discussion look good but the results are very simple. It should be more detailed and stronger.

Line 12, what is the mean line-conditioned excess

Line 155, only S11? Please show the temporal profile at more sites.

[Figure]

Figure. 2c is not described in results.

Figures 2 and 3, why do you choose 2 m? how about 1 or 1.5m?

Figure 3, Missing 0-2m data at 5 sites.

Figure 3, please show the SD as figure 2 did.

Figure 4, how about 0-2m?

Line 485, the variability of precipitation is so large, which makes it not convincing.

Figure 5, please show the SD of x-variable in fig5 b-d.

The results should also include a table with the differences in this method and the other commonly-used isotope methods.

––––––––––––––––––––––––––––

---

## Author Comment (AC1) · 10 Mar 2020

Response to Anonymous Referee #1

We appreciate your positive comments on our manuscript, and expect that we have a chance to revise the manuscript and can address all of the comments raised by the Referee #1. Overall, the two main suggestions were raised and responded as follow:

1. The selection and processing of data sources make it difficult to evaluate the validity of the results. Generally, the introduction and discussion look good but the results are very simple. It should be more detailed and stronger. The results should also include a table with the differences in this method and the other commonly-used isotope methods.

[Figure]

Response: Thank you very much for your suggestions. We will carefully revise our manuscript according to your suggestions. We have already validated our estimates by comparison with the estimates reported by previous studies in the Discussion (section 4.3, L250-260). We will present the estimates from the well-known Craig-Gordon model to further validate our results. Overall, we believe that these issues can be addressed and will further consolidate our main results and conclusions of this manuscript.

2. There are only four sampling sites for precipitation across this area and its variability is so large, which makes it unconvincing.

Response: Thanks for raising your concerns. We are assuming that the Referee concerns the variability of precipitation lc-excess, presented in the Figure 4. In the Figure, we presented the the Boxplot of the mothly precipitation lc-excess from all precipitation sampling sites. We believe that the variability of the precipitation lc-excess values would not originate from the limited precipitation sampling sites for the following reasons.

(1) The four precipitation stations cover the precipitation variability over the region. And all the monthly precipitation data collected from different periods, are closely plotted along a single line ( the regional LMWL), and parameters (slope (from 6.11to 7.56) and intercept (from -7.76 to 7.72)) of LMWL did not vary much among the stations (Figure A1, which will present as the supplementary).

(2) the LMWL is comparable with that of the previous study in this region ($\delta2H$ = 6.90$\delta18O$ + 0.10, n = 277, R2 = 0.93) (Li et al., 2019), and the correlation coefficient of the regression equation is over 90%, suggesting the LMWL is reasonable ($\delta2H$ = 6.89 (0.15)$\delta18O$ -0.16 (1.23), n = 212, R2 = 0.91).

(3) For a single precipitation sample (monthly-based here), its lc-excess values would have a tiny deviation from the zero; however, the average value is zero. Because the LMWL parameters (slope and intercept) is derived from many of the monthly data, the uncertainty in these two parameters would be even smaller (please see in the method

section, L145-150). Therefore, the variability of precipitation lc-excess values would not affect the accuracy of our estimates.

Li, Z., Coles, A. E., and Xiao, J.: Groundwater and streamflow sources in China's Loess Plateau on catchment scale, Catena, 181, 104075, 2019.

Response to specific comments

Line12, what is the mean line-conditioned excess?

Response: We modified as "Our results showed that the deep soil water (2 - 10 m) had a mean line-conditioned excess (lc-excess, which describes the offset of a water sample from the Local Meteoric Water Line in the dual-isotope space) less than zero (-13.1‰ to -3.8‰ at the 15 sites across China's Loess Plateau, suggesting that evaporation effects are preserved in the isotopic compositions of the deep soil water."

Line155, only S11? Please show the temporal profile at more sites.

Response: We only monitored the soil water isotope data at one site temporally to address the temporal variation. Therefore, we do not have temporal data from other sites. And this is consistent with general understanding that deep soil water isotopes are not normally affected by precipitation infiltration and surface evaporation. Therefore, the missing temporal data for remaining sites would not affect our main results and conclusion, and we will address your concerns in the result and discussion sections in the new version.

Figure 2c is not described in results.

Response: We described in the L221-225.

Figures 2 and 3, why do you choose 2m? how about 1 or 1.5m?

Response: The critical assumption in the manuscript is that deep soil water isotopes are free of temporal variation. The precipitation infiltration and soil evaporation are two main factors to the variation of soil water isotopes. However, soil evaporation is limited

to a few to dozens of centimeters and precipitation infiltration depth rarely exceeds two meters at our study area. Therefore, we chose 2 m as the steady depth, which can be verified by the dynamics of soil water isotope at one site (S11). We discussed in the L211-220, and we will discuss it more details in the new version.

Figure 3, Missing 0-2m data at 5 sites.

Response: We added the data.

Figure 3, please show the SD as figure 2 did.

Response: Done.

Figure 4, how about 0-2 m?

Response: We will present the data.

Figure 5, please show the SD of x-variable in fig5 b-d.

Response: Done.

―――――――――――――――――――

**Fig. 1.** The Local Meteoric Water Lines (LMWLs) on the China's Loess Plateau.

---

## Referee Comment (RC2) · Anonymous Referee #2 · 3 Apr 2020

Xiang and colleagues present a novel way to study long-term evaporation losses of infiltrated rainfall by using the isotopic fractionation signal of deep soil waters. The approach with a focus on deep soils is novel and they highlight the opportunity and benefit of such an approach with regard to rainfall partitioning. The manuscript is well structured and mostly well written, and the graphs are mostly informative. The topic is of interest for the HESS community and fits to the journal's scope.

The main issue of the presented study is the lack of information of the isotope ratios in the infiltrating rainfall at the individual sites. While it is unclear how the IAEA rainfall data were used in the study, the uncertainty introduced by not having site-specific rainfall isotope data are not assessed and thus not accounted for. If the actual rainfall isotope ratio differs from the IAEA data derived local meteoric water line, the lc-excess

for the soil water ends up being offset over the entire depth profile. In such case, one would assume negative lc-excess results from transport of partly evaporated shallow soil water into deeper depths. For example, if the intercept of the LMWL is overestimated, the lc-excess values of the soil water will be underestimated. If the slope of the LMWL will be underestimated, then lc-excess values of deeper soil water might be underestimated.

I now see that this aspect has been already addressed in the response to Reviewer 1. If II use some numbers provided in Fig. 2, assuming there is d2H = -68 permill and d18O = -9.2 permill in the subsurface (from reading the graph), the lc-excess could range between -4 and 9 permill depending on the assumption of slope being 6.11 or 7.56. Such uncertainty is not discussed yet in the manuscript.

Further, according to Figure 4, the lc-excess at several sites is within the interquartile range of the rainfall lc-excess. Is rainfall lc-excess related to rainfall intensity? More intense rainfall would probably reach deep soils more likely than less intense rainfall.

Any information about soil moisture is missing. Ratios between infiltrated non-fractionated rain water and fractionated soil water storage are discussed, but remain speculation when no mixing volumes are considered. For now, the authors cannot really provide a process description of how fractionated shallow soil water is transported - without or with only little mixing with rainfall water - to deep depths. Without a description of a hydrological process based on observed data leading to lc-excess < 0 in the subsurface, the interpretation remains speculation.

Specific comments:

25: "f" needs to be clearly defined. Currently not given.

39: typo: field

85: runoff, as surface OR subsurface runoff to streams (no rain reaches the stream) or do you mean "surface runoff"?

93: provide temperature, vacuum, and time applied.

110: It's not clear which of the precipitation sampling stations has been used for constructing the LMWL of which soil sampling site. Please clarify. What are the time

periods of the precipitation sampling? Did this overlap with the soil sampling? What was the sampling frequency of the precipitation sampling? Please provide the LMWL for each study site or was one LMWL used for all the sites?

139: What "annual values" are meant?

186: You miss showing the decrease in evaporation fractionation for your study sites. I suggest to replace the plot of d18O with a plot of lc-excess in Figure 2.

188: What does this mean? Thick soil depths prevent surface soil evaporation? Please rephrase to clarify.

190: In the review by Sprenger et al. (2016) that you refer to, they also presented modeling results assuming no preferential flow and even under such conditions, the evaporation fractionation signal was gone with depth because of the "subsequent mixing of the evaporated soil water with nonfractionated precipitation".

199: This interpretation remains speculation since no information about fractionated soil water storage and mixing with non-fractionated rainfall are shown in the results.

224: Unclear how a ratio can have a residence time and how this residence was derived.

229: What do you mean of evaporation having long residence time? Do you mean the tome water resides in the soil until evaporated or transpired?

261: Extending the data set for northern sites, Sprenger et al. (2018, doi:10.1002/hyp.13135, 2018) found similar to your results that rainfall amount could explain most of the lc-excess dynamics, and not ET.

281: typo: 18O

285: You miss to discuss the uncertainty introduced due to lack of LMWL at the specific site.

473: Please add when was the "Deep profile" taken.

Figure 2: Consider replacing d18O in sub plot (b) with lc-excess, since you refer to evaporation fractionation. Having lc-excess plotted over depth would underline your discussion, while showing very similar d2H and d18O profiles does not provide that much information.

Figure 3: Since lc-excess is such an important measure in your study and you claim that the lc-excess changes over depth, it would be very valuable to see the lc-excess on a second axis in the plots shown in Figure 3.

Figure 4: How about sorting the boxplot by e.g., total rainfall? (Figure 5 shows that there is a relationship there)

---

## Author Comment (AC2) · 13 Apr 2020

Response to Anonymous Referee #2

We sincerely appreciate the Anonymous Referee #2 for your detailed, constructive, and very thoughtful comments on our manuscript. Our point-by-point responses are as follows.

1. The main issue of the presented study is the lack of information of the isotope ratios in the infiltrating rainfall at the individual sites. While it is unclear how the IAEA rainfall data were used in the study, the uncertainty introduced by not having site-specific rainfall isotope data are not assessed and thus not accounted for. If the actual rainfall isotope ratio differs from the IAEA data derived local meteoric water line, the

lc-excess for the soil water ends up being offset over the entire depth profile. In such case, one would assume negative lc-excess results from transport of partly evaporated shallow soil water into deeper depths. For example, if the intercept of the LMWL is overestimated, the lc-excess values of the soil water will be underestimated. If the slope of the LMWL will be underestimated, then lc-excess values of deeper soil water might be underestimated. I now see that this aspect has been already addressed in the response to Reviewer 1. If I use some numbers provided in Fig. 2, assuming there is d2H = -68 permill and d18O = -9.2 permill in the subsurface (from reading the graph), the lc-excess could range between -4 and 9 permill depending on the assumption of slope being 6.11 or 7.56. Such uncertainty is not discussed yet in the manuscript.

Response: To address your concerns, we will add site-specific rainfall isotope data for six of our soil sites that are available. Indeed, the lc-excess method presented in this study requires the site-specific LMWL parameters (i.e. slope and intercept) for each soil sampling site, and it should be more accurately if the site-specific LMWL is used. However, for the method to be widely applicable, a method should not require data that are not widely available. Because LMWL may not be available at each site of interests, we constructed the regional LMWL based on IAEA data from five stations, and use it for all of our sites for evaporation calculation. If the site-specific LMWL is much different from the regional LMWL, this, as you stated, would introduce uncertainty.

To asses this uncertainty, we obtained the site-specific LMWL for six of our soil sites, and compared the evaporation estimates obtained from regional LMWL and site-specific LMWL. Figure 1 showed that the difference between the two methods are very small, with the difference ranging from -3.2% to -2.1%. This suggests that the regional LMWL may be acceptable if site-specific LMWL for a individual soil site is not available. We will add this important part in the method, result, and discussion sections of the revised version.

2. Further, according to Figure 4, the lc-excess at several sites is within the interquartile range of the rainfall lc-excess. Is rainfall lc-excess related to rainfall intensity? More

intense rainfall would probably reach deep soils more likely than less intense rainfall.

Response: Thank you for your comments. Large rainfall events are more likely to be "lighter", while small rainfall events tend to be heavier. Therefore, both $\delta18O$ and $\delta2H$ are negatively correlated to rainfall intensity. However, in the dual space, lc-excess or d-excess is not related to rainfall intensity. No matter what the rainfall intensity is, rainfall lc-excess, by definition, is zero. Notwithstanding individual rainfall events may have nonzero lc-excess values due to variations in moisture sources, air mass trajectories, and cloud processes (Dansgaard, 1964; Allen et al., 2018), such variations are very small compared to $\delta18O$ or $\delta2H$, and are also unimportant because they will be damped as waters from those events are mixed together in land scapes (Allen et al., 2018). As you pointed out, large rainfall events generally infiltrate deeper in the soil profile. Additionally, they also infiltrate faster, having less chance for evaporation fractionation. But during the infiltration processes, the "event" or "new" water mixes with prevent or the soil water that experienced evaporation. Consequently, the infiltrating water carries the evaporation signal (more negative lc-excess) into deep soil. This is also consistent with the "piston" flow model for water and chemical transport in soil. Therefore, the deep soil water would have the evaporation signal and lc-excess of deep soil can be used to calculate evaporation ratio.

Though there is mounting evidence that soil water flow on the China's Loess Plateau is dominated by piston flow (Lin et al. 2006; Xiang et al. 2019; Huang et al. 2017). When preferential flow is dominant flow mechanism, the assumption that deep soil lc-excess keeps the evaporation signal, should be tested vigorously. Currently, we are using simulations to test this hypothesis and others, which is beyond the scope of this study.

Dansgaard, W.: Stable isotopes in precipitation, Tellus, 16, 436–468, 1964.

Allen, S. T., Kirchner, J. W., and Goldsmith, G. R.: Predicting Spatial Patterns in Precipitation Isotope ($\delta2H$ and $\delta18O$) Seasonality Using Sinusoidal Isoscapes, Geophys

Res Lett, 45, 4859-4868, 2018. DOI:10.1029/2018gl077458

Lin, R. F. and Wei, K. Q.: Tritium profiles of pore water in the Chinese loess unsaturated zone: Implications for estimation of groundwater recharge, J Hydrol, 328, 192-199, 2006. DOI:10.1016/j.jhydrol.2005.12.010

Xiang, W., Si, B. C., Biswas, A., and Li, Z.: Quantifying dual recharge mechanisms in deep unsaturated zone of Chinese Loess Plateau using stable isotopes, Geoderma, 337, 773-781, 2019. DOI:10.1016/j.geoderma.2018.10.006

Huang, T. M., Pang, Z. H., Liu, J. L., Ma, J. Z., and Gates, J.: Groundwater recharge mechanism in an integrated tableland of the Loess Plateau, northern China: insights from environmental tracers, Hydrogeol J, 25, 2049-2065, 2017. DOI:10.1007/s10040-017-1599-8

3. Any information about soil moisture is missing. Ratios between infiltrated non-fractionated rain water and fractionated soil water storage are discussed, but remain speculation when no mixing volumes are considered. For now, the authors cannot really provide a process description of how fractionated shallow soil water is transported - without or with only little mixing with rainfall water - to deep depths. Without a description of a hydrological process based on observed data leading to lc-excess < 0 in the subsurface, the interpretation remains speculation.

Response: Thanks. This is a very good question. From a large body of research on the Loess plateau, the piston flow is the dominant transport mechanism in soil. As we stated above, we are using simulations to test our assumption and obtain the process understanding.

Specific comments:

(1) 25: "f" needs to be clearly defined. Currently not given.

Response: Thanks. We revised as "Water loss from soil by evaporation is an essential component of the terrestrial water cycle (Good et al., 2015), and its ratio to the total

precipitation (evaporation/precipitation, f) affects how efficiently precipitation water can be used for ecosystems (Good et al., 2017) because it is often considered as the undesirable component (Kool et al., 2014)."

(2) 39: typo: field

Response: Revised.

(3) 85: runoff, as surface OR subsurface runoff to streams (no rain reaches the stream) or do you mean "surface runoff"?

Response: Thanks. We revised as "surface runoff".

(4) 93: provide temperature, vacuum, and time applied.

Response: Thanks. We added the missing information as "Soil water was extracted via the cryogenic vacuum distillation method (Li-2000, LICA, China; less than 0.2 Pa for system pressure, 95°C for heating temperature, and more than three hours for heating time)".

(5) 110: It's not clear which of the precipitation sampling stations has been used for constructing the LMWL of which soil sampling site. Please clarify. What are the time periods of the precipitation sampling? Did this overlap with the soil sampling? What was the sampling frequency of the precipitation sampling? Please provide the LMWL for each study site or was one LMWL used for all the sites?

Response: Thanks. We have responded in the comment 1 and will be added in the revised manuscript.

(6) 139: What "annual values" are meant?

Response: Thanks. We revised as "We estimated the $\alpha^*$(âŰł) and $\varepsilon$k(âŰł) using the long-term (1981-2018) daily temperature and relative humidity. The minimum and maximum temperature were used to consider the uncertainty associated with the conditions under which evaporation occurs (Allen et al., 2019); Consequently, minimum and max-

imum f were calculated, and we reported the mean and standard deviation values."

(7) 186: You miss showing the decrease in evaporation fractionation for your study sites. I suggest to replace the plot of d18O with a plot of lc-excess in Figure 2.

Response: Done.

(8) 188: What does this mean? Thick soil depths prevent surface soil evaporation? Please rephrase to clarify.

Response: Thanks. We revised as "However, the isotope fractionation signatures in deep soil water is not result of direct surface evaporation, because the evaporation front is generally above the depth of 2 m and thus deep soil water is not under the influence of evaporation."

(9) 190: In the review by Sprenger et al. (2016) that you refer to, they also presented modeling results assuming no preferential flow and even under such conditions, the evaporation fractionation signal was gone with depth because of the "subsequent mixing of the evaporated soil water with nonfractionated precipitation".

Response: Thanks. If one uses single isotope, "subsequent mixing of the evaporated soil water with nonfractionated precipitation" makes the signal disappear" can occur practically. But if one use lc-excess, we think the evaporation signal should not disappear as a result of mixing. As we stated above, we are in the process of using simulation study to obtain a process understanding, which is beyond the scope of this study. In the manuscript, we added this as a limitation.

(10) 199: This interpretation remains speculation since no information about fractionated soil water storage and mixing with non-fractionated rainfall are shown in the results.

Response: Thanks. We add that as a limitation of our study.

(11) 224: Unclear how a ratio can have a residence time and how this residence was

derived.

Response: Thanks. We revised as "Assuming a piston flow mode, the mean pore water velocities ($\theta v$) are estimated as 11 cm yr-1 ($\theta v=((6.1*100))/((2015-1963) )$) and 19 cm yr-1 ($\theta\_v=((9.8*100))/((2015-1963) )$) at S11 and S7, respectively. Therefore, precipitation water infiltrated to 2 m and 10 m below of the soil surface needs a long time, about 11-17 years and 53-85 years, respectively. Under such scenario, soils between 2 m and 10 m comprised of water with the mean age ranged between 14 years and 69 years, and its isotopic compositions preserve the evaporation effects over this period. Thus, the f obtained from the average deep soil water isotopic compositions had a mean residence time of about 28 years."

(12) 229: What do you mean of evaporation having long residence time? Do you mean the time water resides in the soil until evaporated or transpired?

Response: Thanks. as we stated in the response (12), we mean the age of the soil water (the time since the rainfall enters soil to the time at which we took a deep soil core and measured soil water).

(13) 261: Extending the data set for northern sites, Sprenger et al. (2018, doi:10.1002/hyp.13135, 2018) found similar to your results that rainfall amount could explain most of the lc-excess dynamics, and not ET.

Response: Thanks. We will add that in the discussion.

(14) 281: typo: 18O

Response: Thanks. We have revised.

(15) 285: You miss to discuss the uncertainty introduced due to lack of LMWL at the specific site.

Response: Thanks. We will add that in the revised manuscript as we stated in our response to your comment 1.

[Figure]

(16) 473: Please add when was the "Deep profile" taken. Figure 2: Consider replacing d18O in sub plot (b) with lc-excess, since you refer to evaporation fractionation. Having lc-excess plotted over depth would underline your discussion, while showing very similar d2H and d18O profiles does not provide that much information.

Response: Thanks. Done as suggested.

(17) Figure 3: Since lc-excess is such an important measure in your study and you claim that the lc-excess changes over depth, it would be very valuable to see the lc-excess on a second axis in the plots shown in Figure 3.

Response: Thanks. Done as suggested.

(18) Figure 4: How about sorting the boxplot by e.g., total rainfall? (Figure 5 shows that there is a relationship there).

Response: Thanks. Done as suggested.
* * *
Fig. 1. Comparison of evaporation estimates derived from the regional LMWL and the site-specific LMWL.